

# Genome-wide association studies reveal stable loci for wheat grain size under different sowing dates

Yi Hong[1,*], Mengna Zhang[1,*], Zechen Yuan[2], Juan Zhu[1], Chao Lv[1], Baojian Guo[1], Feifei Wang[1] and Rugen Xu[1]

[1] Key Laboratory of Plant Functional Genomics of the Ministry of Education/Jiangsu Key Laboratory of Crop Genomics and Molecular Breeding/Jiangsu Co-Innovation Center for Modern Production Technology of Grain Crops/Joint International Research Laborat, Yangzhou University, Yangzhou, China
[2] Jiangsu Internet Agricultural Development Center, Nanjing, China
[*] These authors contributed equally to this work.

## ABSTRACT

**Background**. Wheat (*Tritium aestivum* L.) production is critical for global food security. In recent years, due to climate change and the prolonged growing period of rice varieties, the delayed sowing of wheat has resulted in a loss of grain yield in the area of the middle and lower reaches of the Yangtze River. It is of great significance to screen for natural germplasm resources of wheat that are resistant to late sowing and to explore genetic loci that stably control grain size and yield.

**Methods**. A collection of 327 wheat accessions from diverse sources were subjected to genome-wide association studies using genotyping-by-sequencing. Field trials were conducted under normal, delayed, and seriously delayed sowing conditions for grain length, width, and thousand-grain weight at two sites. Additionally, the additive main effects and multiplicative interaction (AMMI) model was applied to evaluate the stability of thousand-grain weight of 327 accessions across multiple sowing dates.

**Results**. Four wheat germplasm resources have been screened, demonstrating higher stability of thousand-grain weight. A total of 43, 35, and 39 significant MTAs were determined across all chromosomes except for 4D under the three sowing dates, respectively. A total of 10.31% of MTAs that stably affect wheat grain size could be repeatedly identified in at least two sowing dates, with PVE ranging from 0.03% to 38.06%. Among these, six were for GL, three for GW, and one for TGW. There were three novel and stable loci (4A_598189950, 4B_307707920, 2D_622241054) located in conserved regions of the genome, which provide excellent genetic resources for pyramid breeding strategies of superior loci. Our findings offer a theoretical basis for cultivar improvement and marker-assisted selection in wheat breeding practices.

# INTRODUCTION

One of the world's most staple and widely consumed crops, wheat (*Triticum aestivum* L.) is essential for food security around the world. Wheat breeding has undergone several stages, including disease resistance breeding (*Wang et al., 2020*), semi-dwarf breeding

Corresponding author
Rugen Xu, rgxu@yzu.edu.cn

(*Peng et al., 1999*), and high yield and quality breeding (*Zhang et al., 2022*), as a result, vital gains have been made. Despite the area harvested remaining unchanged, the global wheat yield increased from 27,316 hg/ha in 2000 to 34,919 hg/ha in 2021 (FAO 2022, https://www.fao.org/faostat/en/). In recent years, however, the growth of yield per unit area has gradually slowed down in each wheat-producing country and increasingly narrower genetic basis of wheat germplasms, wheat breeding has entered the climbing stage (*Li et al., 2019*; *Yan et al., 2019*). A key factor in meeting consumption demand will be how to increase production in the face of population growth, urbanization, and climate change.

Grain size is an important factor that directly affects both yield and quality in wheat, which is quantified by grain shape and weight. Increasing grain size is therefore one efficient method to increase yield and improve end-use efficiency of wheat (*Gupta et al., 2020*; *Xin et al., 2020*). Recently, many studies have characterized the wheat grain size and multiple QTLs or genes have been found using different mapping strategies. A QTL (*QTKW.caas-5DL*) for thousand-grain weight was identified and validated in a RIL population of Doumai/Shi4185, which explained 12.5–17.4% of the phenotypic variance and was fine mapped into an approximate 3.9 Mb physical interval on chromosome 5D (*Song et al., 2022*). A QTL cluster with the largest PVE (phenotypic variance explanation) of 21.2% for thousand-grain weight was detected on chromosome 4B (*Yang et al., 2022*). *Qgl1.hau.1B*, a stable QTL for grain length with the PVE of 7.67%–14.45%, was fine mapped into a 0.98-Mb physical interval. The causal gene *TaGL1-B1* encodes carotenoid isomerase and overexpression of which can enhance grain length through interaction with *TaPAP6* (*Niaz et al., 2023*). *KL-PW* is a major grain length gene at the P1 locus in Polish wheat, which encodes a MIKC-type MADS-box protein and significantly increased grain length of wheat (*Chai et al., 2022*). The PGS1 basic helix-loop-helix protein regulates *Fl3* to affect grain development in both wheat and rice. *TaPGS1* overexpression altered expression levels of genes that are related to seed development and increased grain size and weight (*Guo et al., 2022*). In addition, several other genes controlling wheat grain size were isolated by homology cloning, such as *TaGW2* (*Su et al., 2011*; *Lv et al., 2022*), *TaD11-2A* (*Xu et al., 2022*), *TaTGW6* (*Hanif et al., 2016*).

Wheat grain size-related traits are complex quantitative traits that tend to be sensitive to environments, whereas climate is the biggest individual driver contributing to agricultural production variability (*Gupta et al., 2020*). With the changing climate, rainy weather frequently appears in the area of the middle and lower reaches of the Yangtze River, resulting in wheat sowing dates being delayed. Late sowing results in a decrease in wheat yield, with an estimated reduction of 50–60 kg/mu. In cases of severe late sowing, the yield reduction can exceed 30%. The sowing date mainly regulates individual development and population formation by affecting the pre-winter accumulated temperature. On the other hand, late-sown wheat is vulnerable to high temperatures during its late growth stage, especially during filling. As it generally flowers late, this results in the grain filling period overlapping with period of high temperature and water stress, which can drastically reduce the final yield and affect grain quality (*Ahmed & Fayyaz-ul Hassan, 2015*). From economic and environment-friendly viewpoints, therefore, one of the most important objectives of modern wheat breeding programs is to screen wheat genotypes with stable

yields and wide adaptation to diverse environments. In the meantime, identifying loci that stably control grain size under late-sown stressed conditions will also become an important research topic. Stable MTAs identification can provide a foundation for exploring candidate genes, gaining further insights into the molecular mechanisms behind crop yield formation and environmental responses. Additionally, it can offer genetic resources for marker-assisted selection for stable-yield breeding.

Natural populations are rich in genetic resources, making them a powerful tool for identifying natural variations and superior alleles. In this study, a collection of 327 wheat accessions from different sources were used to characterize grain size in three sowing dates at multiple sites. The AMMI model combined with genome-wide association analysis enabled us to screen stable germplasm resources and identified several loci that stably control grain size under late-sown stressed conditions. Our results will provide a theoretical basis for wheat breeding with high and stable yields.

## MATERIALS & METHODS

### Plant materials and field trials

A collection of 327 wheat accessions of Chinese and foreign origin, including cultivars, breeding lines, and landraces, was used in this study (Table S1). To evaluate the performance of wheat grain size and late sowing tolerance (late sowing can make wheat more susceptible to pre-winter freezing damage and high temperature stress during the filling period), field trials were conducted in Yangzhou (YZ, 32°N, 119°E) and Yancheng (YC, 33°N, 120°E) under three different sowing dates, namely stage I (27 October, normal sowing), stage II (10 November, delay), and stage III (24 November, seriously delay). Yangzhou and Yancheng are located in distinct regions of the same wheat ecological zone, both ensuring the normal growth of wheat in practice. The environmental conditions of Yangzhou and Yancheng exhibit some notable differences. Compared to Yangzhou, Yancheng is located closer to the Yellow Sea, with lower average annual temperatures and rainfall. For each sowing date, all accessions were grown in rows 0.6m long and 0.3m apart at a sowing rate of 12 seeds per row according to randomized complete blocks with three replications. Therefore, each repeated block consists of 327 rows with an approximate area of 80 $m^2$. Two sites adopt the same experimental design. Irrigation and management at each site followed local practices.

### Phenotype evaluation and statistical analysis

Five plants with consistent growth of each accession were bulk-harvested and threshed after full maturation. Phenotypic evaluation of wheat grain size, including thousand-grain weight (TGW), width (GW), and length (GL), was conducted on a 200–300 grain subsample using a digital imaging system (WSeen SC-G automatic seed selection and thousand grain weight analysis system). All cracked grains were removed before measurement to exclude trial errors.

SPSS software v21.0 (IBM SPSS, Armonk, NY, USA) was used for descriptive statistical analysis, analysis of variance (ANOVA), and Pearson correlation analysis. Linear mixed models for multi-environment trial (MET) were performed in the R package "lme4" to

obtain the best linear unbiased prediction (BLUP) of grain size under each sowing date by combining field trail data from Yangzhou and Yancheng (*Bates et al., 2015*). Genotype and site were considered random effects.

The AMMI model combines analysis of variance and principal component analysis into a unified approach, incorporating the strengths of both methods. Currently, it has been widely used in regional yield analysis for important crops such as rice, maize, barley, canola, and soybeans. To evaluate the stability of wheat grain size, TGW in six different environments of sites and sowing dates combination was analyzed by AMMI model in the R package "Agricola". Firstly, the significance of genotype-by-environment interactions (GEI) was determined by ANOVA. Then, principal components that reached significant level in interactions (IPCAs) were selected by principal component analysis and their contribution rates were also analyzed. Finally, the stability parameter D (the Euclidean distance between the point of accession and the origin in the principal component space of interactions) was introduced to evaluate the stability of each accession on TGW. The detailed model and formula derivation referred to *Pan et al. (2022)*.

## SNP calling and Genotyping

To genotype this wheat population (327 accessions), genotyping-by-sequencing strategy (*Wallace & Mitchell, 2017*) was performed by Novogene Co., Ltd. Pooled libraries were sequenced using Illumina Hiseq PE150. In total, 601Gb of clean data were obtained, with an average of 1.8Gb for each accession. The average sequencing depth of GBS tags was 7.61 x. After quality control, high-quality reads were aligned to the wheat genome (Chinese Spring genome v2.1) using the Burrows-Wheeler Aligner (*Li & Durbin, 2009*; *Zhu et al., 2021*). In this study, SNPs with minor allele frequency (MAF) $\leq$ 5% and missing rate $\geq$ 20% were excluded. As a result, a total of 69,441 high-quality SNPs were obtained at the genome-wide level.

## Population genetics

To obtain a comprehensive understanding of the population structure in this study, we applied two distinct approaches based on 69,441 SNPs, namely the clustering algorithm ADMIXTURE and principal components analysis (PCA). Admixture clustering of individuals was performed with genetic clusters predefined as $K = 1$ to 10 in the software Admixture (*Alexander, Novembre & Lange, 2009*). Maximum likelihood estimates were generated for each accession derived from each of the K populations, and the cross-validation error (CV_error) was used to identify the best K value. PCA was performed in the software GCTA to characterize the genetic relationships among accessions in this population and validate the divided subgroups from the clustering algorithm ADMIXTURE (*Wang et al., 2021*).

The vcftools software was utilized to measure the nucleotide diversity (Π) of the wheat population for each chromosome and the genetic differentiation index (Fst) between different subgroups, using a sliding-window approach with 10 Mb windows sliding in 5 Mb steps (*Danecek et al., 2011*). Linkage disequilibrium (LD) was estimated as the correlation coefficient ($r^2$) for all pairs of SNPs within 20 Mb using PopLDdecay software
(*Zhang et al., 2019*). LD decay curves for each subgenome were determined by graphing $r^2$ values against the physical distance.

### GWAS for wheat grain size under normal and late-sown stressed conditions

GWAS was performed using a FarmCPU model incorporating population structure in the R program ''rMVP'' (*Yin et al., 2021*). BLUP values of grain size were used as the overall performances of the 327 wheat accessions. The genome-wide significance thresholds ($8.89 \times 10^{-5}$) were determined using a uniform threshold of 1/n, where n is the effective number of independent SNPs calculated using the Genetic type 1 Error Calculator (*Li et al., 2012*). The phenotypic variance explanation (PVE) of SNPs that exceeded the LOD threshold was examined by the R function ''aov()'' and the Pearson correlation coefficient ($r$) between trait values and stable MTAs was also calculated using Excel 2021. MTAs repeatedly detected under at least two individual sowing dates were considered to be stable in this study.

## RESULTS

### Phenotypic variation of wheat grain size under normal and late-sown stressed conditions

Phenotypic variability of the 327 wheat accessions was observed in all the field trials, as showed in a statistical table (Table S2) and frequency distribution plots (Figs. 1A and 1B). Among the three traits related to grain size, TGW had the highest coefficient of variation. Moreover, the coefficient of variation for TGW in YC (17.44%–20.50%) was higher than that in YZ (15.97%–17.43%) under different sowing dates. Late-sown stressed conditions could affect the distribution of each trait. With the sowing date being delayed, the average TGW decreased, but alterations of GL and GW varied across sites. ANOVA results showed significant variations among genotypes, sites, sowing dates, genotype-by-site interactions, and genotype-by-sowing date interactions for all three traits measured (Table 1). TGW, GL, and GW were highly positively correlated with each other and the significances were not affected by sites or sowing dates (Figs. 1C and 1D).

### Stability analysis of TGW

TGW is a crucial indicator of grain yield, as it more directly reflects the final yield compared to GL and GW. The ANOVA was proceeded to look at genotype-by-environment interactions of TGW and the result demonstrated a significant GEI effect, emphasizing the utility of AMMI analysis in identifying stable genotypes. A total of five IPCAs that reached significant level were extracted by principal component analysis. The contribution rate of each IPCA was 31.95% (IPCA1), 28.90% (IPCA2), 17.03% (IPCA3), 13.51% (IPCA4), and 8.61% (IPCA5) (Table 2). Based on the contribution rates and scores of these IPCAs, the stability parameter D was calculated for each accession and subsequently sorted in ascending order (Table S1). As shown on the biplot, D value referred to the distance between accession and origin (Fig. S1). Among the 327 wheat accessions, four accessions with low D values (<0.10) were screened, including Nonglin46, Ningmai23, Ningmai19,
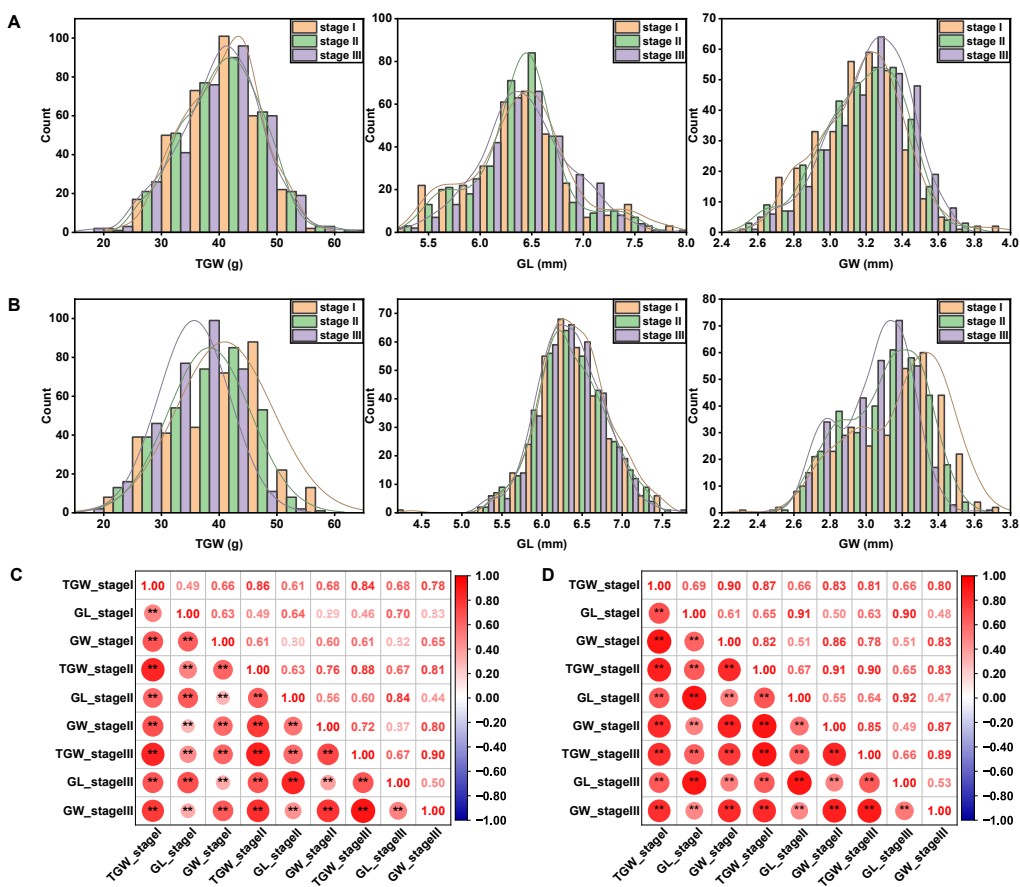

**Figure 1** Frequency distribution and Pearson's correlation coefficients between grain size-related traits under different sowing dates at YZ (A, C) and YC (B, D) sites. Asterisks (* and **) represent the significance at 0.05 and 0.01 level, respectively. YZ and YC represent Yangzhou and Yancheng, respectively. stage I, stage II, and stage III represent normal, delayed, and seriously delayed sowing conditions, respectively. GL, GW, and TGW represent grain length, grain width, and thousand-grain weight, respectively.

and Yangmai3, which exhibited good resistance to late-sown stressed conditions. Interestingly, out of the four screened varieties, three of them originate from the area of the middle and lower reaches of the Yangtze River in China, where the sowing date of wheat is prone to delays due to rainy weather frequently appearing. This outcome suggests that the breeding practices in this region are specifically designed for local conditions, and these main popularized varieties can be used for improvement of late sowing tolerant varieties in other regions.

## Population structure, linkage disequilibrium, and molecular diversity

In this study, all the accessions were initially divided into genetic clusters labeled as K, ranging from 1 to 10, in order to elucidate the population structure. A notable inflection point at $K = 2$ was clear in the cv_error graph, and cv_error values decreased slowly without any discernible valleys as the K increased, suggesting that genetic exchanges

**Table 1  Analysis of variance for wheat grain size.**

| Source | df | Mean of squares TGW | | |
|---|---|---|---|---|
| | | TGW | GL | GW |
| Genotype | 326 | 753.45** | 2.84** | 0.71** |
| Site | 1 | 6631.95** | 2.40** | 9.25** |
| Sowing date | 2 | 3961.51** | 0.39** | 1.30** |
| Genotype × Site | 326 | 37.19** | 0.12** | 0.06** |
| Genotype × Sowing date | 652 | 22.02** | 0.12** | 0.03** |
| Site × Sowing date | 2 | 2841.02** | 0.98** | 4.23** |
| Genotype × Site × Sowing date | 652 | 22.05** | 0.11** | 0.04** |
| error | 3880 | 1.82 | 0.05 | 0.02 |

Notes.
** The significance at 0.01 level.

**Table 2  PCA of genotype-by-environment interactions of TGW.**

| Source | df | Mean of squares | Contribution rates (%) |
|---|---|---|---|
| Genotype × Environment | 1630 | 25.10** | |
| IPC1 | 330 | 38.49** | 31.95 |
| IPC2 | 328 | 35.02** | 28.90 |
| IPC3 | 326 | 20.76** | 17.03 |
| IPC4 | 324 | 16.57** | 13.51 |
| IPC5 | 322 | 10.63** | 8.61 |

Notes.
** The significance at 0.01 level.

occur frequently within this population (Fig. 2A). At $K = 2$, individuals were divided into two relatively separated subgroups based the degree of artificial selection, which was also reflected by discrete clusters in the PCA plot (Figs. 2B and 2C). Subgroup 1 comprised nearly all landraces, whereas subgroup 2 mainly consisted of modern cultivated varieties (Table S1). At $K = 3$, the landrace subgroup remained unchanged, while finer subdivisions could be observed within the subgroup of modern cultivated varieties, which to some extent corresponded to agro-ecological regions (Figs. 2B and 2D).

Genome-wide linkage disequilibrium analysis revealed the slowest LD decay and longest LD distance for the A subgenome, while the D subgenome exhibits the opposite trend (Fig. 2E). The mean Fst estimate between subgroup 1 (landraces) and subgroup 2 (cultivars) was 0.28 and high genetic differentiation could be observed at whole genome level. In comparison to the D genome with a Π value of 2.89E-07, A (2.42E-06) and B (2.32E-06) genomes showed extensive nucleotide diversity (Fig. 3). Notably, certain segments of the genome displayed low degree of genetic differentiation among subgroups and overall genetic diversity (especially on the D subgenome), indicating that these segments may contain a significant number of repetitive sequences or have been relatively conserved during the process of selection and domestication. In contrast, other segments showed a higher degree of differentiation among subgroups despite low overall nucleotide
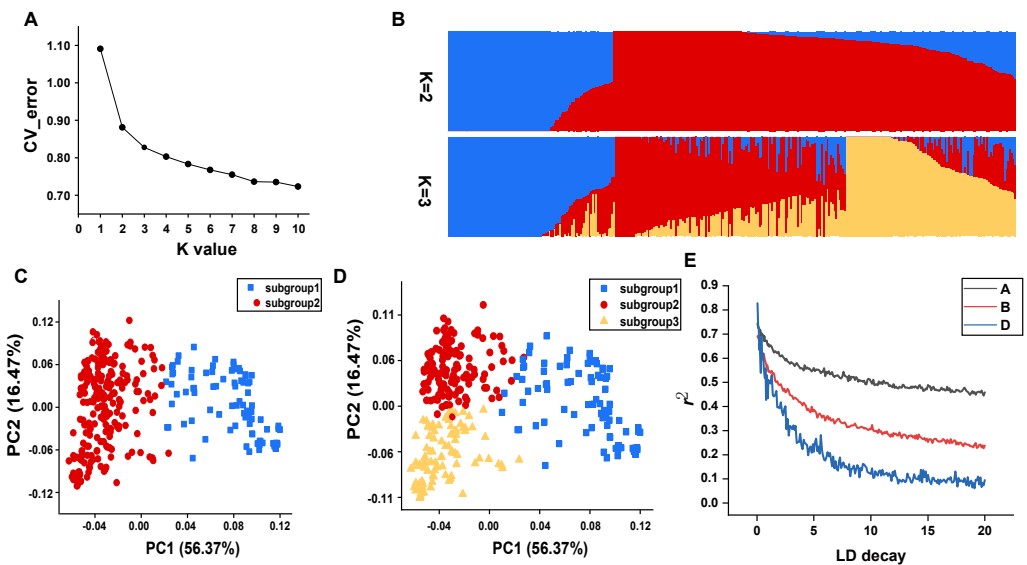

**Figure 2  Population genetic characteristics.** (A) Cross-validation error plot ($K = 1$ to $K = 10$). (B) Population structure of the 334 accessions at $K = 2$ and $K = 3$. (C) The PCA plot demonstrates the division of population into two subgroups. (D) The PCA plot demonstrates the division of population into three subgroups. (E) The linkage disequilibrium decay plots for each subgenome.

diversity (at 100–150 Mb on chromosome 3A), suggesting that strong selection occurred during wheat breeding in these regions.

## Marker-trait associations

Genome-wide association studies of wheat grain size identified 43, 35, and 39 MTAs under three sowing dates, respectively, across all chromosomes except for 4D in this study. The repeatedly identified MTAs include a set of MTAs that were detected under late-sown stressed conditions, showing consistency with normal-sown conditions, as well as a set of MTAs that were specifically and repeatedly detected under late-sown stressed conditions. We referred to these MTAs as stable loci.

Under normal sowing condition (stage I), a total of 43 significant MTAs were detected to associate with grain size, including 16 for GL, 13 for GW, and 13 for TGW (Fig. S2, Table S3). For GL, the range of LOD value for each MTA was between 4.05–6.94. Seven MTAs exhibited PVE of less than 1%, indicating minor effects. However, three MTAs located on the 2D and 3D chromosomes exhibited major effects, explaining 13.10% (2D_587607700), 11.71% (2D_622241054), and 13.49% (3D_170207472) of the PVE, respectively. For GW, the PVE ranged from 0.01% to 22.80%, with five MTAs displaying significant major effects, explaining 22.80% (1B_587916353), 12.88% (3A_152168836), 12.07% (3B_487351200), 19.14% (5A_691891912), and 17.10% (6D_450824747) of the PVE, respectively. For TGW, most of the significant MTAs detected exhibited major effects, except for 2D_309856702 (1.57%), 3B_459942548 (3.50%), 5B_688598683 (0.01%), and 6A_584257945 (9.71%).

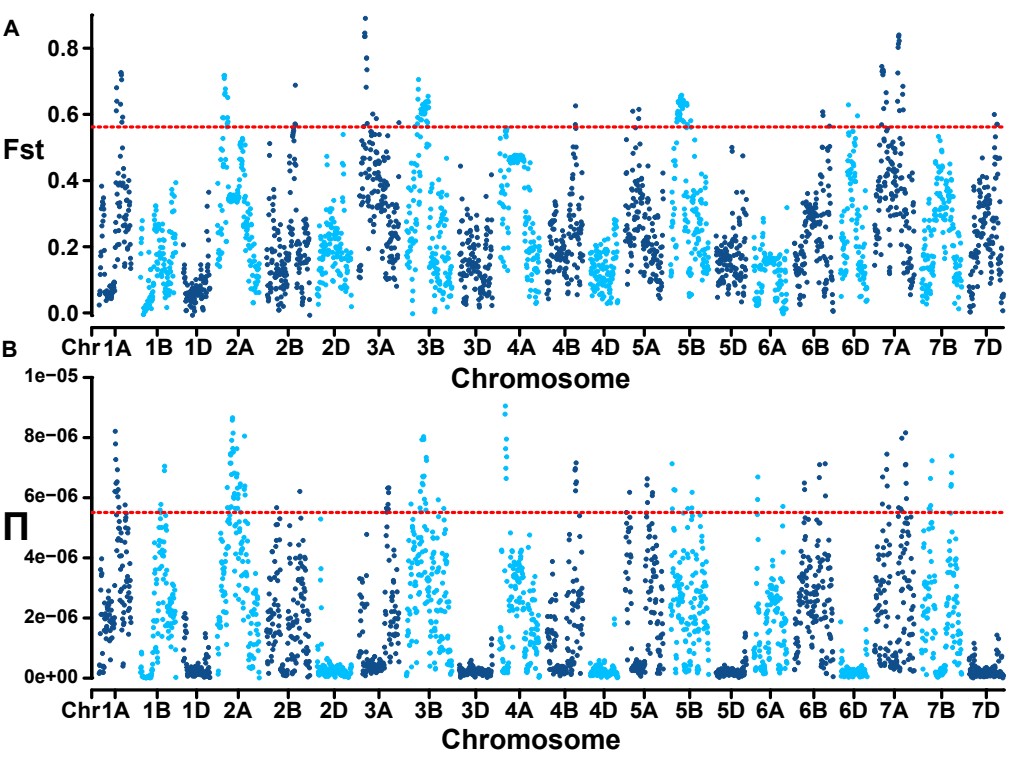

**Figure 3** **Population divergence (Fst) across the two subgroups (A) and nucleotide diversity (Π) of wheat population at whole genome level (B).** Red lines mark the genome-wide threshold at the top 5%.

Under delayed sowing condition (stage II), a total of 35 significant MTAs were detected that are related to grain size, with 13 MTAs affecting GL, 9 MTAs affecting GW, and 13 MTAs affecting TGW (Fig. S3, Table S4). For GL, the LOD values of each significant MTA ranged from 4.29 to 14.75, with PVE ranging from 0.30% to 21.03%. Among them, six MTAs explained a relatively high PVE, namely 2D_622241054 (12.92%), 4A_598189950 (15.86%), 5B_473824350 (12.51%), 6B_619025969 (13.34%), 6B_661282224 (21.03%), and 7D_524516821 (17.27%). For GW, the LOD range of MTAs is between 4.03–9.76, out of which four MTAs exhibited major effects and accounted for 24.99% (1A_405294165), 10.04% (2B_610553852), 16.80% (4B_622203495), and 38.06% (5A_31907623) of PVE, respectively. As for TGW, most loci had PVE less than 10%, and only three MTAs located on chromosomes 4A, 6A, and 7A exhibited major effects, accounting for 18.55% (4A_22859078), 10.09% (6A_584257945), and 18.70% (7A_607804063) of PVE, respectively.

Under seriously delayed sowing condition (stage III), a total of 39 significant MTAs were detected, with 13 each for GL, GW, and TGW (Fig. S4, Table S5). As for GL, the PVE of each MTA ranged from 0.23% to 21.66%. Among them, six MTAs had major effects, including 1B_558317639 (14.50%), 2A_712168308 (13.00%), 2D_622241054 (12.94), 4A_598189950 (14.79%), 5A_451968303 (15.71%), 6B_661282224 (21.66%). For GW, five MTAs showed major effects and could explain PVE of 15.13% (1B_466412697),

**Table 3  Repeatedly identified SNP for grain size under normal- and late-sown stressed conditions.**

| Traits | SNP | Chromosome | Position(bp) | Ref | Alt | Sowing dates | LOD | PVE(%) | Correlation |
|---|---|---|---|---|---|---|---|---|---|
|  | 2A_612223880 | 2A | 612223880 | T | C | Stage I and III | 4.98–5.98 | 0.26–0.54 | 0.05–0.07 |
|  | 2D_622241054 | 2D | 622241054 | G | T | Stage I, II, and III | 6.94–14.75 | 11.71–12.94 | 0.34–0.36 |
| GL | 4A_598189950 | 4A | 598189950 | T | C | Stage II and III | 5.44–5.46 | 14.79–15.86 | (−0.38)–(−0.40) |
| | 4B_307707920 | 4B | 307707920 | G | A | Stage I, II, and III | 4.05–5.58 | 5.93–7.06 | 0.24–0.27 |
|  | 6B_661282224 | 6B | 661282224 | A | G | Stage II and III | 5.10–6.80 | 21.03–21.66 | (−0.46)–(−0.47) |
|  | 7A_13137024 | 7A | 13137024 | A | G | Stage II and III | 4.52–4.63 | 1.53–1.72 | 0.12–0.13 |
|  | 4A_678723323 | 4A | 678723323 | G | A | Stage I and II | 4.10–4.66 | 2.43–2.56 | 0.15–0.16 |
| GW | 5A_31907623 | 5A | 31907623 | A | G | Stage II and III | 4.81–5.50 | 35.86–38.06 | (−0.60)–(−0.62) |
|  | 5B_42550541 | 5B | 42550541 | C | T | Stage II and III | 4.03–8.74 | 0.03–0.72 | 0.02–0.08 |
| TGW | 6A_584257945 | 6A | 584257945 | G | A | Stage I, II, and III | 4.20–8.68 | 9.71–10.10 | 0.31–0.32 |

**Notes.**

A minus sign (−) preceding correlation coefficient indicates negative correlation.

25.70% (1B_682158610), 35.86% (5A_31907623), 11.59% (7B_658023779), and 29.70% (7D_524516821) respectively. For TGW, the LOD of MTAs ranged from 4.12 to 11.84. Six MTAs could explain over 10% of PVE, namely, 2B_11347661 (10.08%), 2D_316459408 (11.14%), 5B_541942664 (11.26%), 6A_584257945 (10.10%), 6B_552167191 (10.09%), and 7A_620653950 (15.92%).

Based on the consistent identification of linkage SNP, multiple loci can be repeatedly identified at two or three sowing dates (Table 3). The correlation coefficient ($r$) between genotype and phenotype ranged from 0.05 to −0.47 (the minus sign denotes a negative correlation). Stable loci for GL located on chromosomes 2A, 2D, 4A, 4B, 6B, and 7A. Among these, 4B_307707920 was detected in all cases and exhibited stable $r$ values, despite not being the major effector. Furthermore, four loci were exclusively detected under late-sown stressed conditions. Interestingly, three of them exhibited $r$ values exceeding 0.35. Stable loci for GW located on chromosomes 4A, 5A, and 5B. 4A_678723323 and 5B_42550541 had lower $r$ values under the first two and last two sowing dates, respectively, while 5A_31907623 had higher $r$ values under the last two dates. Only a single locus was repeatedly detected to be associated with TGW in all cases, which had stable $r$ values and showed major effects under late-sown stressed conditions. The identification of these stable MTAs provides preliminary genomic positions for the subsequent exploration of genes related to wheat adaptation to the challenging environment.

# DISCUSSION

## Effects of sowing dates on wheat growth and development

In recent years, the extended growth period of rice varieties and the frequent occurrence of rainy weather have resulted in a progressively delayed sowing period of wheat in eastern China, especially across mid-lower reaches of the Yangtze River (*Han et al., 2019*). Effects of sowing dates on the growth and development of wheat are primarily manifested in the pre-winter freezing damage and the high-temperature-forced ripening during the grain-filling stage. As wheat grows, it exhibits degrees of sensitivity to temperatures during
different growth stages (*Slafer & Rawson, 1995*). If wheat is sown on schedule, a sudden spring cold may result in freezing damage to the vigorously growing wheat seedlings with high accumulated temperature before winter (*Basheir et al., 2023*). During the grain-filling stage, heat stress will result in forced ripening, thereby a decrease of grain filling duration, total aboveground biomass, and grain yield (*Liu et al., 2016*). In this study, we established three sowing dates with a maximum time gap of approximately one month to ensure that the test materials were exposed to different light, temperature, water, and air conditions at various growth stages. The result showed that with the sowing date being delayed, the average TGW decreased, but alterations of GL and GW varied across sites. As described by *Zhang et al. (2013)*, the wheat cultivars Jingdong8 and Lunxuan987 experienced a 5-day delay in sowing, resulting in yield reductions of 9.2% and 10.6% respectively. With a 10-day delay in sowing, both varieties experienced a further decrease in yields by 3.8%. Moreover, there was a significant decrease in TGW of wheat. Another study also demonstrated that delaying the sowing date has varying impacts on plant height and grain yield for 15 popularized wheat varieties in the Huanghuai region (*Wang et al., 2022*). ANOVA results showed significant variations among genotypes, sowing dates and genotype-by-sowing date interactions for wheat grain size. These findings align with the previous study conducted by *Wang et al. (2021)*. Grain size-related traits are complex quantitative traits that are largely influenced by environments. In this study, the trial sites (Yangzhou and Yancheng) are located in different regions with varying conditions such as light, temperature, and water. Therefore, significant effects were detected at the trial sites. By using the AMMI model, four accessions with stable TGW were screened, which can be exploited for crop improvement.

## Comparison of the stable MTAs to known QTL

In this study, a total of 117 significant MTAs were detected to associate with wheat grain size under three sowing dates. Most of these loci were expressed as an environment-dependent pattern, which were identified under single sowing condition. However, we have still detected several highly stable MTAs, with the maximum number for GL and the least for TGW. We further compared their physical positions with known QTL identified previously. *Miao et al. (2022)* refined 394 initial TGW QTL into 67 MQTLs through the integration of individual maps from 45 studies using Meta-QTL analysis. Here, the stable and major MTA (6A_584257945) for TGW at 584 Mb on chromosome 6A was discovered within the genomic region of MQTL-6A-6 (535–593 Mb). The stable MTAs for GL on chromosomes 2A (2A_612223880) and 6B (6B_661282224) were co-located with MQTL-2A-4 (509-613Mb) and MQTL-6B-5 (603-713Mb), respectively. The stable MTAs for GW on chromosomes 4A (4A_678723323) and 5B (5B_42550541) were co-located with MQTL-4A-4 (666–679 Mb) and MQTL-5B-2 (10–80 Mb), respectively. Although these MQTLs were from TGW, co-detections revealed hotspot genomic regions that control wheat grain size, owing to the strong correlation among these traits. In another Meta-QTL analysis conducted by *Saini et al. (2022)*, 89 MQTLs were found to associate with grain morphology-related traits. Among these, MQTL5A.3 located within 11–40.3 Mb on chromosome 5A and overlapped with MTA 5A_31907623. As

for MTA 7A_13137024, although it was beyond the genomic region of MQTL7A.1, the overlapping confidence interval suggested that they were likely to be the same locus. On chromosome 2D, 4A, and 4B, no previously reported MQTLs neighbored the GL-related MTAs 2D_622241054, 4A_598189950, and 4B_307707920, respectively. They could potentially be novel discoveries.

## Candidate genes of the stable and novel MTAs

Genes are genetic factors that regulate phenotype, and identifying the causal gene of MTA will facilitate genetic analysis of important agronomic traits in crops. In this study, we revealed three novel and stable MTAs for GL under late-sown stressed conditions. MTA 4A_598189950 was detected under stage II and III, it was able to account for 15.86% and 14.79% of PVE, with high correlation coefficients −0.40 and −0.38, respectively, between genotype and phenotype. At this locus, *TaRGB1* (*TraesCS4A03G0746100*) was an obvious candidate gene which is orthologous to rice *RGB1* and encodes the heterotrimeric G protein $\beta$-subunit (G $\beta$). G protein signaling pathway is one of the important pathways for regulating seed size in model crops. Knock-down of *RGB1* result in short seeds due to reduced cells number of hull, suggesting that *RGB1* positively regulate seed length (*Utsunomiya et al., 2011*). Otherwise, the presence of *RGB1* serves as the basis for the impacts of G $\gamma$ on controlling rice seed size (*Li, Xu & Li, 2019*; *Ren, Ding & Qian, 2023*). Spikelet hulls undergo maturation before wheat grains begin filling, determining the scale of the cavity in which the integuments produce the seed coat. Both spikelet hull and seed coat have an impact on ultimate grain morphology, including grain length (*Hong, Zhang & Xu, 2023*). Referring to the WheatOmics database (http://wheatomics.sdau.edu.cn/), it appears that *TaRGB1* highly expressed in spike, which raises the possibility that it may be a candidate gene for GL-related MTA 4A_598189950.

MTA 4B_307707920 neighbors the centromeric region, within which highly repetitive sequences affect genes' fine mapping and strong linkage disequilibrium suppresses recombination frequencies (*Su et al., 2019*; *Hong, Zhang & Xu, 2023*). This locus had no obvious candidate gene. At the end of chromosome 2D, MTA 2D_622241054 was repeatedly detected under all sowing conditions and demonstrate major effects. In previous study, GWAS and knockdown experiments have revealed the role of *TaARF12* (on chromosome 2A) in regulating wheat plant height and grain yield (*Li et al., 2022*). It is worth noting that mutations in rice *ARF12* also led to short GL and decreased TGW (*Qiao et al., 2021*), suggesting its conserved functional mechanism across species. Here, we have identified *TaARF12-2D* (*TraesCS2D03G1217700*), a homolog to *TaARF12*, which is a promising candidate for MTA 2D_622241054 on chromosome 2D and is highly expressed in the stem and spike according to WheatOmics database.

In future research directions, conducting haplotype analysis and expression level detection on the candidates mentioned above will enable rapid determination of the correlation between phenotype and genotype, while also facilitating the acquisition of superior haplotypes. Furthermore, with the mature application of gene editing technology in wheat, there is potential for uncovering the underlying molecular mechanisms of these candidates.

### Breeding utilization and marker-assisted selection

Wheat has always been a focus of breeders in terms of its yield and quality. In recent years, marker-based mapping approaches have become important means of revealing associations between target traits and genetic regions (*Hong, Zhang & Xu, 2023*). In this study, three novel GL-related loci were found to show stability in normal- and late-sown stressed conditions. These loci are located within conserved regions of the wheat genome, with MTA 4B_307707920 neighboring the centromeric region, while 2D_622241054 and 4A_598189950 are in regions of low genetic differentiation and nucleotide diversity. Notably, prolonged natural selection and artificial domestication may have caused certain loci or regions to undergo selection sweeps, which is consistent with a previous study (*Li et al., 2022*). Unsurprisingly, we found that *TaTB1*, a wheat domestication gene related to inflorescence architecture, is located in selection sweeps and near 4A_598189950. Studies have shown that altering the dosage or function of *TaTB1* can help increase wheat yield (*Dixon et al., 2018*). Thus, these novel and stable loci possess a higher value for breeding purposes.

The novel and stable loci can be screened using linkage markers to assist in selecting plants with excellent grain size characteristics. In practice, marker-assisted selection can greatly shorten the breeding cycle and reduce unnecessary human, material, and financial inputs. At the same time, it can also help breeders select stable and high-yielding wheat varieties, thereby improving agricultural production efficiency. Hence, the identification of loci associated with wheat grain size through GWAS, coupled with marker-assisted selection, will provide new ideas and methods for wheat breeding, making greater contributions to global food security.

## CONCLUSION

The occurrence of yield losses in wheat production due to delayed sowing, influenced by multiple factors, necessitates the identification and cultivation of wheat varieties that exhibit tolerance towards late-sown stressed conditions. Additionally, the exploration of yield-related loci that demonstrate consistent performance across diverse environments represents an effective and eco-friendly strategy. This study successfully identified ten loci that consistently influence wheat grain size across multiple sowing dates using GWAS analysis. It is worth noting that three of these loci were previously unidentified. Furthermore, four wheat germplasm resources were identified as suitable candidates for breeding purposes, as they demonstrated high stability in terms of thousand-grain weight under late-sown stressed conditions, as determined by AMMI analysis. These findings provide a solid basis for the development of wheat varieties with both high yield and those adapted to late sowing.

### Funding

This work was supported by the Jiangsu Province seed industry revitalization project (JBGS2021006), the National Key Research and Development (2017YFD0100803), and a Project Funded by the Priority Academic Program Development of Jiangsu Higher Education Institutions, China. The funders had no role in study design, data collection and analysis, decision to publish, or preparation of the manuscript.

### Grant Disclosures

The following grant information was disclosed by the authors:
Jiangsu Province seed industry revitalization project: JBGS2021006.
National Key Research and Development: 2017YFD0100803.
Project Funded by the Priority Academic Program Development of Jiangsu Higher Education Institutions, China.

### Competing Interests

The authors declare there are no competing interests.

### Author Contributions

- Yi Hong analyzed the data, prepared figures and/or tables, authored or reviewed drafts of the article, and approved the final draft.
- Mengna Zhang analyzed the data, prepared figures and/or tables, authored or reviewed drafts of the article, and approved the final draft.
- Zechen Yuan performed the experiments, prepared figures and/or tables, and approved the final draft.
- Juan Zhu performed the experiments, prepared figures and/or tables, and approved the final draft.
- Chao Lv performed the experiments, prepared figures and/or tables, and approved the final draft.
- Baojian Guo performed the experiments, prepared figures and/or tables, and approved the final draft.
- Feifei Wang performed the experiments, prepared figures and/or tables, and approved the final draft.
- Rugen Xu conceived and designed the experiments, prepared figures and/or tables, authored or reviewed drafts of the article, and approved the final draft.

### DNA Deposition

The following information was supplied regarding the deposition of DNA sequences:
The 327 wheat accessions high-throughput sequencing data are available at SRA: PRJNA1044234.

### Data Availability

The raw measurements are available in the Supplementary Files.
## Supplemental Information

Supplemental information for this article can be found online at http://dx.doi.org/10.7717/peerj.16984#supplemental-information.

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
