# Peer review of "Genome-wide association studies reveal stable loci for wheat grain size under different sowing dates"

_PeerJ, doi:10.7717/peerj.16984_

## Round 0.1 · original submission · Major Revisions

The manuscript underwent revision based on feedback from two reviewers who recommended further changes. Please incorporate these suggestions and resubmit the revised version.

**Language Note:** The review process has identified that the English language must be improved. PeerJ can provide language editing services - please contact us at copyediting@peerj.com for pricing (be sure to provide your manuscript number and title). Alternatively, you should make your own arrangements to improve the language quality and provide details in your response letter. – PeerJ Staff

·

Basic reporting

The manuscript needs some changes

Experimental design

The study demonstrates an acceptable experimental design.

Validity of the findings

No comments

Additional comments

I reviewed the paper titled "Genome-wide association studies reveal stable loci for wheat grain size under different sowing dates". The study provides valuable insights for improving wheat cultivars and implementing marker-assisted selection in breeding practices. It highlights the importance of germplasm resources and stable genetic factors in achieving consistent grain size.
-Comments and Suggestions for Authors
Abstract
The abstract is well written.
- It would be even more impactful to include the percentage of stable loci identified.
- Consider highlighting the number and potential significance of the three novel loci to further emphasize the study's contribution to the field.
Introduction
- Mentioning the estimated yield reduction due to late sowing would further emphasize the problem's significance.
- Mentioning the unique aspect of identifying MTAs for stable grain size under late-sown stress would further highlight the study's contribution.
Materials and methods
The Materials & Methods section is well-written and provides a clear overview of the research design and methodologies employed. The inclusion of diverse plant material, multiple environments, appropriate statistical analyses, and genotyping strengthens the study's validity and generalizability.
- Briefly explain the rationale behind choosing specific statistical models (e.g., AMMI for stability) would enhance transparency.
- While the overall population size is large, providing details about the sample size used for GWAS would strengthen the power and generalizability of the identified MTAs.
- Clarifying how MTAs were identified as stable across multiple sowing dates would be helpful.
Results
-The results section is well written.
- The authors should add further emphasis on the most impactful findings (e.g., major MTAs, stable genotypes); this could be beneficial for readers seeking a quick overview.
- Please mention the potential applications or implications of the identified stable genotypes and MTAs for improving wheat resilience. This could add further value to the research.
Discussion
-The discussion section is comprehensive and well written.
- The discussion could be slightly more concise, especially in sections describing previously known QTLs.
- The authors should mention specific future research directions (e.g., functional validation of candidate genes), this could be a valuable addition.
Conclusion
-This conclusion effectively summarizes the key findings of the study and emphasizes their practical significance for wheat breeding. It clearly highlights the identification of stable loci and germplasm resources, making it easy for readers to grasp the impact of the research.

Reviewer 2 ·

Basic reporting

The manuscript entitled "Genome-wide association studies reveal stable loci
for wheat grain size under different sowing dates" is dealing with an important issue, which is how to avoid the negative effect of late sowing date on the performance of wheat through searching for stable loci related to grain size that can consistently maintain grain yield under different environments.
The paper was written in understandable English, however some English editing are required. (Please see comments on Attached document).

Experimental design

More details must be provided in the section of experimental design and analyses to clarify how the experiment was carried out and how the analyses were processed.
Some points must be clarified like including grain yield in the study, using two experimental sites in terms of abiotic stresses, selection of stable marker-trait association...

(Please see comments on the attached document for more details).

Validity of the findings

no comment

Additional comments

Please see comments on the attached document and respond and revise the document point by point.

Annotated reviews are not available for download in order to protect the identity of reviewers who chose to remain anonymous.

---

## Round 0.2 · accepted · Accept

The manuscript has undergone review by two reviewers and both have recommended its acceptance.

·

Basic reporting

no comment

Experimental design

no comment

Validity of the findings

no comment

Additional comments

The authors have made the changes I suggested in the last review. I recommend its publication in this journal.

Reviewer 2 ·

Basic reporting

no comment

Experimental design

no comment

Validity of the findings

no comment

Additional comments

no comment